# The Effect of Servant Leadership on Work Engagement: The Role of Employee Resilience and Organizational Support

**DOI:** 10.3390/bs14040300

**Published:** 2024-04-05

**Authors:** Mingyue Cai, Minghui Wang, Jiajia Cheng

**Affiliations:** School of Economics and Management, Nanjing Tech University, Nanjing 211816, China; 202121013001@njtech.edu.cn (M.C.); 202221013062@njtech.edu.cn (M.W.)

**Keywords:** servant leadership, employee resilience, organizational support, work engagement

## Abstract

Employees suffer from low resources in the workplace because of multiple work roles in project-based organization (PBO). Based on the conservation of resources theory (COR), this study identifies both employee resilience and organizational support as critical personal and job resources. It then examines how servant leadership enhances employee work engagement in PBO through the mediating roles of employee resilience and organizational support. This study uses a questionnaire-based quantitative research design to collect data from 437 employees in PBO. The collected data were analyzed by partial least squares structural equation modeling (PLS-SEM) to test hypotheses. The research findings indicate that servant leadership positively affects work engagement. Additionally, the relationship between servant leadership and work engagement is mediated by employee resilience and organizational support. This study deepens the understanding of how servant leadership promotes work engagement in PBO by providing personal and job resources. The findings deepen our understanding of how servant leadership enhances work engagement in PBO. The findings also provide implications for PBO to enhance sustainable well-being at work and organizational productivity.

## 1. Introduction

The most productive and competitive organizations are composed of valuable employees who are physically, cognitively, and emotionally engaged in their work [1]. Particularly following the deployment of the United Nations Sustainable Development Goals (UNSDGs), motivating work engagement has become a prime concern of many organizations [2]. PBO is no different in this regard. Nevertheless, due to the majority of the business conducted in terms of projects, PBO faces challenges in motivating work engagement [3,4,5]. PBO has been extensively used to conduct business in projects in some industries, such as construction, information technology, and manufacturing [6]. In such organizations, employees undertake multiple work roles across different projects simultaneously, which may cause work stress and thus impair work engagement, which is an essential aspect of well-being [7,8].

As a result, employees need to obtain different resources to deal with the high work demands and maintain well-being in a stressful scenario [9]. However, based on the job demands–resources theory (JD-R), employees show the best job performance in work environments that combine the challenged job demands with job resources because such environments facilitate their work engagement [10]. Bakker and Demerouti (2007) argued that employees with high personal and job resources are more likely to harness high work demands and achieve elevated well-being [11]. Therefore, personal resources and job resources are used in this study to enhance work engagement in the workplace to achieve a sustainable well-being-productivity synergy.

According to the conservation of resources theory (COR), individuals strive to acquire new resources or reserve their valued resources to defend against the potential threat of resource loss after substantive investment [12]. Thus, employees with high job and personal resources are more likely to deal with stressful tasks caused by the complex, dynamic, and uncertain characteristics of projects [13], leading to high-level well-being in the workplace [9]. Specifically, personal resources enable employees to overcome stress and cultivate positive self-evaluations related to resilience [14]. As a valuable personal resource, employee resilience describes a kind of positive psychological capabilities to cope with adversity, ambiguity, conflict, and failure, as well as to shoulder positive events, progress, and increasing duties [15]. In PBO, employee resilience describes the positive psychological capacity to withstand stressful work situations. For example, employee resilience allows employees to recover efficiently from project experiences with tight schedules and high complexity and then devote themselves to their work [16]. Consequently, employee resilience becomes a strong predictor of work engagement. Compared with personal resources, job resources (e.g., organizational support) have an equally positive effect on work engagement [17]. According to Bakker and Demerouti (2007), job resources are social, organizational, and physical resources that lessen workloads, make goals easier to achieve, and foster personal development [11]. Organizational support describes a general employee’s perception that the organization values their contributions and cares about their well-being and development [18]. Organizational support focuses on the positive aspects of organizational features like concern, acknowledgment, and respect for employees. When employees perceive their efforts as being recognized and rewarded by their organization, they are motivated to exert their utmost effort into their work.

Previous studies have advocated that reasonable job resources such as organizational support, job autonomy, and personal resources reflected in aspects like self-efficacy or resilience are positively related to engagement [11]. Following the JD-R theory, organizational and personal resources act as a buffer against demands while helping employees stay engaged [17]. Therefore, it is important to maintain a delicate balance that goes beyond the dichotomy of resources versus demands in order to maintain engagement when faced with the demanding tasks of a project. Although personal and job resources are crucial predictors of work engagement, generating various resources depends on positive leadership behavior [19]. It is noticeable that servant leadership, as a follower-centric leadership style, provides proper resources to employees with such care that can be useful in enhancing the vigor, dedication, and absorption of employees in a project context [20]. This is primarily because servant leadership focuses on employees’ development and needs, which is likely to be overlooked in PBO due to the mentality of profit maximization [21]. These things considered, servant leadership establishes a supportive work environment to decrease stressful situations and helps employees achieve project goals [22]. According to the COR theory, servant leadership helps establish coping mechanisms and enables employees to minimize valued resource losses under a stressful situation [23]. As a result, employees can respond to the dynamic and stressful project context and complete project tasks within a limited time.

This research aims to investigate the impact of servant leadership on work engagement by examining the mediating role of employee resilience and organizational support. It contributes to the current literature in the following ways. First, this research highlights the significant role of servant leadership, an effective organizational intervention, in enhancing work engagement within a project context. Second, our research emphasizes the importance of resource generation functions of servant leadership in project management. Third, shedding light on the COR theory and JD-R theory, this research identifies the employee resilience (personal resources) and organizational support (job resources) that mediate the effect of servant leadership on work engagement. The research helps project leaders effectively enhance work engagement by accumulating reasonable resources, ultimately improving sustainable well-being at work.

## 2. Literature Review and Hypotheses

### 2.1. Work Engagement in PBO

Kahn (1990) identified cognitive, emotional, and physical aspects as three dimensions of work engagement [24]. Furthermore, Schaufeli et al. (2002) defined work engagement as “a positive, fulfilling, work-related state manifested by vigor, dedication, and absorption in work roles” (p. 74) [25]. Accordingly, work engagement involving affective connection and commitment influences how individuals behave. Engagement is the motivation of employees to work hard and their desire to leave no stone unturned to achieve organizational goals.

Work engagement positively correlates with organizational success and yields multiple positive outcomes. Prior evidence indicated that work engagement has a significant influence on job and task performance, productivity, organizational citizenship behaviors, discretionary efforts, emotional commitment, and customer service [26,27]. It reflects employees’ willingness to undertake organizational tasks. Work engagement is also associated with higher profit levels, overall revenue generation, and growth [17]. Therefore, recent studies have concentrated on enhancing employee work engagement as a strategy for improving performance. In PBO, projects are utilized to achieve organizational goals, necessitating that project leaders focus on enhancing employee work engagement and aligning it with organizational needs [28]. Thus, work engagement plays a critical role in promoting project performance.

Some extrinsic factors that could enhance work engagement include culture and purpose, growth opportunities, overall compensation, quality of life, job characteristics, leadership, and interpersonal relationships [29]. In particular, job characteristics, organizational support, salary, and supervisory constitute a set of job resource variables that significantly predict engagement [30]. Boamah and Laschinger (2015) verified a correlation between psychological capital and increased work engagement [31]. As mentioned in the JD-R theory of motivational processes, job resources have motivational potential and lead to high work engagement, low levels of cynicism, and excellent performance [32]. Accordingly, job resources are significant predictors of work engagement and have underscored the increasing importance of leadership development programs in fostering work engagement [33].

Hakanen et al. (2006) proposed that job resources constitute the most significant predictors of work engagement [34]. Consequently, work engagement is often attributed to job resources [33]. Resources can be categorized into two categories: (a) the individual level, encompassing personal resources, and (b) the organizational level, including leadership styles and organizational practices [17]. Furthermore, Bakker et al. (2007) suggested that job resources significantly influence motivation particularly when job demands are elevated [10]. Thus, due to the high-demand environment in PBO, employees need to acquire substantial resources to cope with such an environment [35].

### 2.2. Servant Leadership and Work Engagement

A servant leadership style refers to leaders whose primary purpose for leading is to serve others by investing in their development and well-being for the benefit of completing tasks and objectives for a common goal [36]. This leadership helps subordinates grow to reach their maximum potential for optimal organizational and professional success and places the needs of his or her subordinates before his or her own [37]. Thus, servant leadership emphasizes the significance of prioritizing employees’ comfort and growth as a top priority. In contrast to conventional leadership, servant leadership asserts that serving others constitutes a leader’s primary responsibility. While previous research has shown that servant leadership can positively impact a variety of team outcomes, it is unclear which of these leadership behaviors accurately fosters a supportive and sustainable work environment. Accordingly, given the temporary and dynamic attributes of projects, servant leadership requires further exploration in PBO.

Servant leadership is crucial in enhancing program performance and achieving the long-term objectives of PBO [38]. Previous studies have demonstrated that servant leadership positively influences key aspects of organizational management, such as employee performance [39], team effectiveness [40], and positive workplace behavior [41]. Thus, servant leadership fosters the positive behaviors of participants that enhance job effectiveness [42]. Servant leadership also places significant emphasis on dimensions such as altruism, ethics, empathy, and, most importantly, prioritizing the needs of followers [43,44]. Addressing and serving the needs of followers to spur growth constitutes the foundation of the servant leader paradox [22,45]. This kind of leadership style ensures that the needs of followers are met in addition to fostering their personal development [46].

In PBO, the insufficiency of work resources is a primary factor influencing work engagement [47]. According to the COR theory, work resources (e.g., organizational support and employee resilience) act as both intrinsic and extrinsic motivators, enhancing employee learning, development, and growth [48]. Similarly, the demonstration of positive effort and engagement at work is contingent upon an individual’s resources [49]. Studies have indicated that developing and maintaining a well-resourced organizational environment is effective in mitigating concerns regarding resource scarcity [50,51]. Servant leaders play a pivotal role in shaping an environment and behaviors conducive to providing resources. Sousa and Dierendonck (2017) asserted that servant leadership has the potential to assist employees in overcoming the negative effects associated with inadequate resources [22]. In other words, servant leadership mitigates a lack of engagement stemming from concerns about their work resources. Servant leaders inspire their followers by providing resources and prioritizing dedication to their development, listening, compassion, and persuasion, and thus encouraging them to serve and oversee subordinates [52].

According to the COR theory, resources are believed to motivate and enhance job devotion [48]. Accordingly, employees may derive benefits from the servant leadership behaviors of their managers, potentially inspiring them to grow and develop on a personal level and intrinsically motivating them through acts of selflessness, thereby fostering greater job dedication [50,53]. Consequently, we hypothesize that servant leaders will elevate work engagement by fostering an environment that nurtures employees’ motivation to complete tasks. Thus, the following hypothesis is proposed:

**H1.** *Servant leadership is positively related to work engagement in PBO*.

### 2.3. Servant Leadership and Organizational Support

Organizational support is an important work resource that reflects the organizational concern for the contributions, welfare, and interests of employees [54]. Employees engage in highly demanding or overloaded roles, such as those within PBO, and require organizational support to flourish in stressful work environments. Assurance of being valued and cared for by the organization not only boosts their self-esteem but also shields them from the harmful effects of stress. JD-R theory proposes that job resources buffer the effects of job demands on stress [55]. In other words, organizational support mitigates work-related stress and work fatigue, potentially preventing depression [56]. Subsequently, employees tend to exhibit greater loyalty and dedication alongside higher levels of job satisfaction [54]. Thus, providing support and fair treatment to employees is not only a moral manner but a sound business strategy. Conversely, employees lacking organizational support tend to experience greater job dissatisfaction, are more prone to conflicts with colleagues, and generally exhibit reduced confidence [57].

Given the disadvantageous of lacking organizational support, Zhang et al. (2012) contended that servant leadership is a crucial source of organizational support [53]. With such leadership, leaders exhibit proactivity, follower-centricity, and self-sacrifice by prioritizing the needs of their followers above both their interests and those of the organization [22,58]. Employees thus positively engage in their work upon perceiving opportunities for personal and professional growth and development, coupled with receiving management support [59]. Consequently, they may invest emotion, dedication, and passion in their work, stemming from personal and professional development [30,59]. Thus, the subsequent hypothesis is formulated:

**H2.** *Servant leadership is positively related to organizational support in PBO*.

### 2.4. Servant Leadership and Employee Resilience

Employees possess personal resources, which they can leverage alongside organizational resources, to manage work-related responsibilities [60]. In high-stress environments in PBO, employee resilience constitutes a crucial personal resource characterized by an individual’s capacity to recover from stress [61]. Employees with greater psychological resilience demonstrate higher levels of work engagement, being more adept at navigating adverse circumstances and sustaining a positive perspective on their profession [62]. Existing research has explored how to bolster employee resilience in a negative environment. For example, Kuntz et al. (2017) argued that supportive organizational elements, such as the cultivation of servant leadership, play an important role in fostering employee resilience [63]. Wu and Lee (2017) asserted that positive leadership qualities aid team members in cultivating positive psychological resources [64]. Specifically, employees experience beneficial psychological effects and demonstrate increased motivation to complete tasks when they can easily access abundant resources at work [65,66]. By effectively utilizing psychological resources, individuals proactively prepare for challenges and mitigate the impact of stressful events on themselves.

Given that the preventive mitigation of risks or stressors is frequently challenging in PBO, resource-centered responses should be strengthened. Servant leaders, serving as resource providers in the workplace environment, contribute to building employee resilience to enhance work engagement. Consequently, the following hypothesis is proposed:

**H3.** *Servant leadership is positively related to employee resilience in PBO*.

### 2.5. The Mediating Role of Employee Resilience

In positive psychology, employee resilience is defined as positive coping and adaptation when confronted with significant risk or adversity, which is crucial in the workplace [67]. Previous research considered that employee resilience represents a positive psychological ability to recover from adversity, uncertainty, conflict, failure, or even positive change, progress, and increased responsibility [64]. According to Parker et al. (2015), employee resilience is an important personal resource that encompasses those psychological assets or resources that enable individuals to thrive and prosper in a positive workplace [68]. The resilience of frontline employees in projects allows them to maintain composure in stressful situations and assist in protecting their emotional resources [68]. Thus, employee resilience serves as a reasonable mediator. With higher levels of employee resilience, there is a greater impact of servant leadership on work engagement in PBO.

With regard to employee resilience, Wiroko (2021) argued that servant leadership serves as a crucial antecedent that fostering employee resilience to motivate high-level work engagement [69]. Competent and nurturing leaders are able to provide sufficient personal and social resources to their followers to enhance employee psychological capital such as employee resilience [70]. Subsequently, their followers will possess positive self-assessment and thus believe that they are qualified to deal with uncertainty and stress [71]. This illustrates that sufficient work resources can promote work engagement. Referring to COR theory, employees with a lack of personal resources (low-level employee resilience) may be unable to withstand adversity or actively adapt in the workplace [48]. As a result, employees who exhibit low-stress resilience tend to lose motivation to work, thereby reducing the impact of a servant leadership style on work engagement. In contrast, when employees exhibit high resilience in their work roles, they perform given tasks effectively and exhibit higher levels of work engagement [72]. Hence, the following hypotheses are formulated:

**H4.** *Employee resilience has a positive effect on work engagement in PBO*.

**H5.** *Employee resilience plays a mediating role in servant leadership and employee work engagement in PBO*.

### 2.6. The Mediating Role of Organizational Support

Organizational support means that the organization values the contributions of its employees and cares about their well-being and development [73]. Eisenberger et al. (1986) argued that employees’ perceived organizational support develops within the context of considering organizational behavior as a whole, and it precisely reflects the overall support perceived by employees at the organizational level [73]. Empirical studies have demonstrated that followers’ work engagement positively correlates with employees’ perceptions of organizational support [74,75].

The impact of servant leadership on work engagement is a resource-based motivational process in which employees receive organizational support through a series of interactions that are integral to the motivational process [32,76], culminating in positive performance [77]. In PBO, servant leaders motivate employees to improve their performance by providing organizational support to their followers and meeting their needs. However, PBO employees often lack adequate organizational support while facing increased stress. In addition, due to a strong sense of resource conservation, employees are less motivated to accomplish the organization’s mission, thus discouraging them from being fully engaged and immersed in their work. However, the core goal of the servant style is to prioritize the needs of employees and provide work resources. According to COR theory, job resources are the main motivating factor [48]. So, when employees feel that the organization appreciates their contributions and cares about their well-being, they tend to be more motivated to go above and beyond and achieve more. In contrast, when employees lack organizational support, the scarcity of organizational resources leads to a strong desire to protect, which will further reduce work engagement in PBO. Therefore, the following hypotheses are developed:

**H6.** *Organizational support has a positive effect on work engagement in PBO*.

**H7.** *Organizational support plays a mediating role in servant leadership and employee work engagement in PBO*.

## 3. Materials and Methods

### 3.1. Sample and Procedures

This study collected quantitative data by designing a questionnaire. The questionnaire is divided into two parts: the first part includes a demographic survey of the respondents and the second part includes measures of the independent variable (servant leadership), the dependent variable (work engagement), and two mediating variables (organizational support and employee resilience). All items were rated by a 5-point Likert scale from “1 = strongly degree to 5 = strongly agree”.

Items were derived from the literature and incorporated into the project-based organization context. Since the questionnaire was quoted from the previous studies, to ensure the appropriateness and accuracy of the translation, we firstly sent the questionnaire to some eligible target respondents to fill in and evaluate and revised some of the text according to the feedback. After that, an anonymous survey was conducted.

The data were collected from October 2022 to January 2023. This study used a snowball sampling method by asking respondents to provide other relevant respondents to obtain a large amount of data. The targeted respondents were project leaders or members working on different projects, particularly in the manufacturing, IT, and construction sectors in which the PBO form is prevalently utilized to conduct business and production activities [6]. In this survey, PBO is a kind of permanent organization containing multiple temporary units (projects) [3]. Their engaged PBO conducted the majority of their activities in project mode with some functional support and coordination [4]. Although the respondents are moving from one project to another or from project mode to organizational roles, they are mostly affiliated to organizations with permanent employment contracts [78,79].

To increase the response rate of respondents, this study clarified the purpose, content, meaning, instructions for completion, and confidentiality of the survey in the questionnaire. Additionally, web-based and paper-based questionnaires were distributed to the respondents. Ultimately, 437 electronic questionnaires were received and carefully examined. Therein, some of the questionnaires answered in a short time were invalid. These things considered, some of the questionnaires that were not answered by the participants that engaged in the PBO form required by this study were invalid as well. As a result, 81 were found to be invalid, thereby reducing the number of valid questionnaires to 356, with a total effective response rate of 81.46%.

In addition, of the respondents, 24.71% were from construction, 33.64% were from information technology, 29.29% were from manufacturing, and the remaining 12.36% were from others. The one-way ANOVA was adopted to compare each variable among respondents engaged in construction, manufacturing, information technology, and other industries. There was no significant difference in servant leadership (F (3, 352) = 1.209, *p* = 0.306 > 0.05), organizational support (F(3, 352) = 0.631, *p* = 0.596 > 0.05), employee resilience (F (3, 352) = 0.320, *p* = 0.811 > 0.05), and work engagement (F(3, 352) = 0.639, *p* = 0.590 > 0.05). Therefore, the one-way ANOVA did not reveal any statistically significant differences in the findings.

The basic information of the respondents is exhibited in Table 1.

### 3.2. Measurements

This study measured the variables by a five-point Likert scale from “strongly disagree = 1” to “strongly agree = 5”. The specific measures are exhibited in Table 2.

Servant leadership was measured by a 7-item scale from Nauman et al. (2022) [80]. In the research of Nauman et al. (2022), servant leadership functions in a project context to serve the team and motivate a collective synergy for goal attainment. The Cronbach’s alpha was 0.903. The items involve the perception of participants on leadership effectiveness, such as “In the last completed project, my leader put my best interests ahead of their own”.

Employee resilience was measured using 5 items developed by Al-Hawari et al. (2020) to underline the importance of personal resources which enable employees to exhibit resilient behaviors when facing challenges [81]. A sample item is “I can get through difficult times at work because I have experienced before difficulties”. The value of Cronbach’s alpha for employee resilience is 0.891.

Organizational support was measured using a 4-item scale developed by Cheng et al. (2003) [82]. The 4-item scale of organizational support can effectively examine employee outcomes in the workplace, such as job satisfaction, work engagement, and turnover intention. The items mainly described the perceived support from their organizations. The sample items include “My organization won’t take advantage of me whenever there’s an opportunity”. The value of Cronbach’s alpha for organizational support was 0.878.

Work engagement was measured using a 9-item scale developed by Schaufeli et al. (2006) [83]. The 9-item scale has good psychometric traits and can be applied to examine employee engagement for positive organizational behavior research, which has been widely used in the project context [20,84]. A sample item is described as “I am immersed in my work”. The Cronbach’s alpha was 0.937.

## 4. Results

### 4.1. Measurement Model

Internal consistency reliability was tested using composite reliability (CR) and Cronbach’s Alpha. Table 2 indicates that the values of Cronbach’s Alpha and the CR of each construct exceeded the 0.7 benchmark and 0.8 benchmark, which evidences that there are appropriate levels of internal consistency reliability.

Indicator reliability was tested by factor loading. Table 2 shows that all the factor loadings are more than the suggested 0.70 value, indicating acceptable item reliability [85].

This research utilized the average variance extracted (AVE) to test the convergent validity of the survey instrument. As exhibited in Table 2, each of the AVE values (0.632–0.732) exceeds the 0.5 criterion. The results confirm that the convergent validity was acceptable.

The Heterotrait–Monotrait (HTMT) ratio was used to evaluate discriminate validity. Table 3 shows that all HTMT values were less than 0.85, proving that the constructs’ discriminant validity was satisfactory [86,87].

### 4.2. Common Method Variance

This research used questionnaire-based surveys to collect data, which may cause common method variance. Accordingly, this research employed Harman’s single-factor test as a statistical remedy to assess common method variance. The results exhibit that the first factor explained 43.625% of the variance, which was less than 50% [88]. Hence, the common method variance is not a serious issue in this research.

### 4.3. Structural Model

This research built a complete PLS-SEM structural model (Figure 1) to assess the hypotheses involving all the research variables. The model’s predictive power was tested by the coefficient of determination R^2^ for the endogenous constructs. Figure 1 exhibits that the R^2^ values (0.239, 0.210, and 0.239) are over 0.20, indicating a satisfactory predictive power [89]. The cross-validated redundancy index (Q^2^) was applied to examine the predictive relevance of the structural model. Each Q^2^ value of the endogenous constructs is higher than 0.2 (in Figure 1), which indicates a predictive appropriateness of the model [86].

Results in Figure 1 demonstrate that servant leadership is positively correlated with work engagement (H1: β = 0.314, t = 6.094, *p* < 0.001). Servant leadership is positively correlated with employee resilience and organizational support (H2: β = 0.489, t = 11.753, *p* < 0.001, H3: β = 0.458, t = 10.393, *p* < 0.001), which implies that the positive effect of servant leadership on employee resilience and organizational support designers is significant. Organizational support and employee resilience are positively correlated with work engagement (H4: β = 0.318, t = 6.326, *p* < 0.001; H5: β = 0.214, t = 4.136, *p* < 0.001), indicating that employee resilience and organizational support have a positive impact on work engagement.

As H2 and H4 are supported, the survey formulates a mediated hypothesis that reveals the mediating effect of employee resilience between servant leadership and work engagement. As Figure 1 shows, the direct effect of servant leadership on work engagement is significant (β = 0.314, t = 6.094, *p* < 0.001), while the indirect effect through employee resilience is also significant (β = 0.155, t = 5.701, *p* < 0.001). Thus, employee resilience plays an indirect mediating role between servant leadership and work engagement.

As H3 and H5 are supported, the survey formulates a mediated hypothesis that reveals the mediating effect of organizational support between servant leadership and work engagement. As Figure 1 shows, the direct effect of servant leadership on work engagement is significant (β = 0.314, t = 6.094, *p* < 0.001), while the indirect effect through organizational support is also significant (β = 0.098, t = 3.934, *p* < 0.001). Thus, organizational support plays an indirect mediation role between servant leadership and work engagement.

## 5. Discussion

An engaged workforce is a potential source of organization-wide competitiveness and strategic advantage. Prioritizing employee well-being will benefit continued health and well-being at work, ultimately leading to high-level work engagement. More specifically, a sustainable workforce effectively reduces the costs associated with burnout, absenteeism, and stress. However, PBO faces the challenge of increasing employee engagement due to insufficient organizational and personal resources. Employees are under pressure to meet strict deadlines and costs, and they do not have sufficient resources to solve difficulties. This study tries to understand how the impact of servant leadership on work engagement based on the COR theory. The study further suggests that personal and job resources can explain how servant leadership affects work engagement in PBO. The research results demonstrate the mediating role of organizational support as a job resource and employee resilience as an individual resource in this link.

This study shows that servant leadership is positively related to work engagement. Past research has concluded that though the specific drivers of work engagement vary by job type, occupational sector, and organization, work engagement peaks when employees are faced with positive events, especially when they are also provided with adequate work resources [28]. Based on past research, our findings show that it is difficult for traditional leaders to increase employee work engagement because PBO is different from traditional organizations in that the completion of projects requires high-level engagement to meet tight schedules. Servant leadership plays an important role in PBO characterized by the high demand for a well-resourced work environment and shaping the role of the servant leader to mitigate low levels of employee engagement and inadequate contributions [20]. When employees work in a well-resourced work environment, they may feel more empowered and valued, resulting in a higher level of work engagement. Since servant leadership can make a significant contribution to employee performance, it is believed to be an effective organizational intervention that promotes social sustainability within the workplace.

Our findings confirm the positive impact of servant leadership on organizational support and employee resilience. The findings illustrate that servant leadership provides organizational support and promotes employee resilience in PBO [90,91]. Consistent with previous findings, leaders play an important role in increasing employee resilience by clarifying work tasks and mentioning support for subordinates. In particular, servant leaders emphasize the personal development of their followers rather than their personal or organizational interests [91]. Servant leaders provide and develop resources in the form of competencies for employees, which employees will seek to retain in the form of resilience. When servant leaders give support to their subordinates, employee resilience is developed, which will increase their ability to cope with stress.

The positive effects of employee resilience and organizational support on work engagement indicate that employee resilience and organizational support as crucial resources motivate employee work engagement in PBO. The COR theory suggests that having sufficient resources can inspire workers. Based on the COR theory, our findings indicate that employee resilience and organizational support, as both personal and job resources, can significantly increase work engagement. Employee resilience is an important personal resource to maintain lasting passion for their work. Our results highlight that job and personal resources are related positively to work engagement. The results echo Xanthopoulou et al. (2009), who argued that the higher the employee resilience, the higher the level of work engagement [17]. Additionally, organizational support maintains good working conditions for employees. Thus, employees can focus more on getting their work done to achieve project goals.

Finally, this study found the mediating role of employee resilience and organizational support. The findings indicate that servant leaders provide sufficient resources in the workplace to motivate employees to proactively engage in work. The findings suggest a significant mediating effect of organizational support and employee resilience on the relationship between servant leadership and employees’ work engagement. A servant leadership style can be used to build and sustain organizational resources with which employees feel that their feedback is respected and appreciated. Thus, this study deepens the understanding of how employees can achieve sustainable well-being at work in the context of limited resources to contribute to the UNSDGs. Specifically, sustainable organizational support motivates employees to participate and work toward a common goal. The results deepen the understanding of how resources affect work engagement. Until now, little has been known about how to promote employee engagement in such a high-demand environment as a project-based organization. This study examines the mediating mechanisms between servant leadership and work engagement from a resource-based perspective. The research results show that organizational support and employee resilience are key resources that predict work engagement and thus help to rationalize the positive correlation between these constructs [60]. This study contributes to the understanding of the potential impact of servant leadership on work engagement by identifying organizational support and employee resilience as factors mediating this critical relationship, thus helping to extend the COR theory.

## 6. Implications

This study aims to illustrate how servant leadership motivates positive work behaviors by providing employees with different work resources. This study extends the results of servant leadership [51], particularly in PBO [60]. Considering the multiple work roles and work stress that erode employee well-being, this study underlines the significance of people-oriented servant leadership in the project management practices of PBO. By clearly defining how servant leadership functions in PBO, this study proposes an effective and positive management intervention to prioritize values, support, and care. Furthermore, based on COR and JD-R theory, this study identifies two urgent work-related resources by which project leaders influence positive work states and the behaviors of employees. The two resources are employee resilience (personal resources) and organizational support (job resources). Both of these work resources emerge as vital mediators between servant leadership and work engagement. The findings extend the research for the antecedents of work engagement and explain how to enhance work engagement effectively through these two work resources.

This study provides some beneficial guidance for managers and decision makers in PBO. Despite PBO facing low employee engagement due to its temporary nature and limited resources, employees may exhibit higher levels of work engagement in the presence of servant leaders. Organizations recognize the critical role of servant leaders in motivating work engagement. Therefore, PBO focuses on providing people-oriented training to develop and promote service behaviors in project managers. In addition, when hiring project managers and project team members, PBO should gauge the service orientation of candidates.

Organizational interventions aimed at providing servant leadership to enhance organizational support and employee resilience in the workplace could prove valuable in increasing work engagement. Thus, project managers concentrate on demonstrating such empowering, supportive, and people-oriented behaviors for the duration of the project. For instance, they can offer organizational support to employees across various dimensions, including the human environment, work system processes, compensation, and benefits. Developing a relatively comfortable and convenient work environment, alongside designating clear performance appraisal standards, ensures that PBO employees are fully engaged in their work. Additionally, leaders acknowledge good work, commend employees for their express gratitude for their efforts, supply appropriate resources and developmental opportunities, offer regular formal or informal feedback (organizational support), and foster a supportive atmosphere [92]. As a result, employees with a higher perception of organizational support actively reduce the negative impact of resource scarcity on their work engagement through a positive mindset, which is key for enhancing sustainable employee well-being.

Project managers implement a series of measures to enhance the work dedication of employees to achieve relevant UNSDGs and social sustainability in the workplace. Examples include cultivating employee resilience by enhancing their professional knowledge and competencies to cope with stressful work environments. Simultaneously, greater attention should be given to developing employee resilience to bolster their psychological capital against work pressure. In addition, the implementation of employee assistance programs, targeted solution models, positive thinking reduction therapy, short-term introspection therapy, and therapeutic introductions can foster positive hope and optimism, encouraging individuals to concentrate on their work and cultivate a higher level of work dedication.

## 7. Limitations and Future Research

Though these findings are noteworthy, there are some limitations. First is the research design. The sample utilized in this research was confined to a single country, China, potentially limiting the generalizability of our results across cultural contexts. Future research could validate the findings with more diverse and cross-cultural samples. The purpose of this cross-sectional study is to explore the mechanisms by which servant leadership influences work engagement in PBO, but it is not clear whether there is a causal relationship between each variable. The relationship among different variables is not a direct causal relationship but is associated with a certain causal sequence, which needs to be further confirmed in prospective long-term studies with a time-lag design.

Besides the limitations in the research design, some unanswered questions need further examination. Although organizational support and employee resilience are identified as urgent work resources in PBO, other types of significant resources are suggested to enhance employee work engagement in the future. For example, social capital is an organizational resource embedded in the relational network [93]. And self-efficacy is another vital personal resource that enables employees to carry out tasks to realize specific objectives [94]. Therefore, future research is suggested to examine the important roles of those work resources in improving work engagement in PBO.

## Figures and Tables

**Figure 1 behavsci-14-00300-f001:**
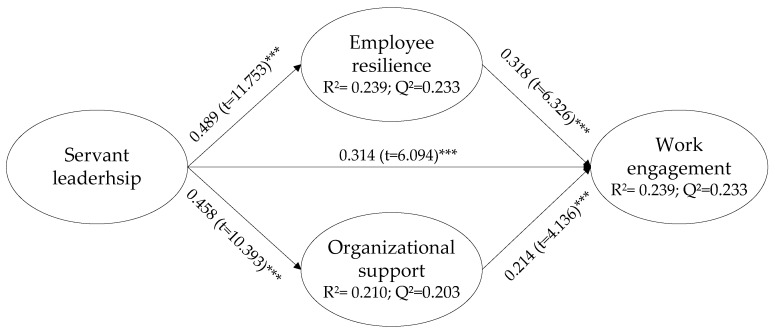
Results of the structural model. Note: *** *p* < 0.001.

**Table 1 behavsci-14-00300-t001:** Demographics and profile of respondents (N = 356).

Variable	Category	Number	Percentage
Gender	Male	210	59.04%
	Female	146	40.96%
Age	Below 25	62	17.39%
	26–35	133	37.30%
	36–45	103	29.06%
	Above 45	58	16.25%
Education level	Below junior college	138	38.90%
	Undergraduate	182	51.26%
	Postgraduate	29	8.01%
	PhD	7	1.83%
Industry	Construction	88	24.71%
	Manufacturing	104	29.29%
	IT	120	33.64%
	Other	44	12.36%
Position	General employees	232	65.22%
	Line manager	68	19.22%
	Middle manager	33	9.15%
	Senior manager	23	6.41%
Work experience	0–5	66	18.54%
	6–10	109	30.66%
	11–15	104	29.06%
	>15	77	21.74%

**Table 2 behavsci-14-00300-t002:** Measurements and reliability.

Constructs and Items	Factor Loading	CR	Cronbach’s a	AVE
Servant leadership		0.923	0.903	0.696
1: In the last completed project, my leader could tell if something work-related was going wrong.	0.743			
2: In the last completed project, my leader made my career development a priority.	0.733			
3: In the last completed project, I would seek help from my leader if I had a personal problem.	0.732			
4: In the last completed project, my leader emphasized the importance of giving back to the community.	0.758			
5: In the last completed project, my leader put my best interests ahead of their own.	0.748			
6: In the last completed project, my leader gave me the freedom to handle difficult situations in the way that I felt was best.	0.697			
7: In the last completed project, my leader would not compromise ethical principles to achieve success.	0.736			
Employee resilience		0.920	0.891	0.732
1: I usually manage difficulties one way or another at work.	0.748			
2: I can be “on my own,” so to speak, at work if I have to.	0.777			
3: I usually take stressful things at work in my stride.	0.781			
4: I can get through difficult times at work because I have experienced before difficulties.	0.778			
5: I feel I can handle many things at a time at my job.	0.754			
Organizational support		0.916	0.878	0.632
1: My organization strongly considers my goals and values.	0.821			
2: My organization cares about my well-being.	0.794			
3: My organization won’t take advantage of me whenever there’s an opportunity.	0.809			
4: My organization shows great concern for me.	0.762			
Work engagement		0.947	0.937	0.665
1: At my work, I feel bursting with energy.	0.774			
2: At my job, I feel strong and vigorous.	0.753			
3: I am enthusiastic about my job.	0.744			
4: My job inspires me.	0.745			
5: When I get up in the morning, I feel like going to work.	0.753			
6: I feel happy when I am working intensely.	0.724			
7: I am proud of the work that I do.	0.747			
8: I am immersed in my work.	0.762			
9: I get carried away when I am working.	0.740			

**Table 3 behavsci-14-00300-t003:** Discriminant validity of the HTMT ratio criterion.

	ER	OS	SL	WE
ER				
OS	0.471			
SL	0.540	0.511		
WE	0.612	0.540	0.612	

## Data Availability

The data presented in this study are available on request from the corresponding author.

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
