# Peer review of "The Effect of Servant Leadership on Work Engagement: The Role of Employee Resilience and Organizational Support"

_behavsci, 2024, doi:10.3390/bs14040300_

Round 1
Reviewer 1 Report
Comments and Suggestions for Authors
The introduction is very broad and well-established in the literature. However, it is worth devoting some space to the JD-R theory, as you have taken the resource part from this theory.
Describe in detail where you took the questionnaire items from and why these were chosen and not others. This especially applies to work engagement and organisational support.
Industries differ, e.g. construction and IT. Did this affect the results? It is worth investigating and describing this in the paper.
Author Response
Reviewer 1 comments
1. The introduction is very broad and well-established in the literature. However, it is worth devoting some space to the JD-R theory, as you have taken the resource part from this theory.
Response: Thanks for your recognition of this study. According to your comments, the authors have instilled JD-R theory into the introduction as follows:
“However, based on the job demands–resources Theory (JD-R), employees show the best job performance in work environments that combine the challenged job demands with job resources because such environments facilitate their work engagement.” (In the second paragraph)
“Following the JD-R theory, organizational and personal resources act as a buffer against demands while helping employees stay engaged.” (In the fourth paragraph)
Reference:
Bakker, A.B.; Hakanen, J.J.; Demerouti, E.; Xanthopoulou, D. (2007). Job Resources Boost Work Engagement, Particularly When Job Demands Are High. Journal of educational psychology, 99, 274.
Xanthopoulou, D.; Bakker, A.B.; Demerouti, E.; Schaufeli, W.B. (2009). Reciprocal Relationships between Job Resources, Personal Resources, and Work Engagement. Journal of Vocational Behavior, 74, 235–244.
2. Describe in detail where you took the questionnaire items from and why these were chosen and not others. This especially applies to work engagement and organizational support.
Response: Thanks for your comments. The authors describe the details of the resource of the questionnaire items and the reasons to choose them as follows, which can also be found in the sub-chapter “3.2 Measurements” section.
Servant leadership was measured by a 7-item scale from Nauman et al. (2022). In the research of Nauman et al. (2022), servant leadership functions in a project context to serve the team and motivate a collective synergy for goal attainment. The Cronbach’s alpha was 0.903. The items involve the perception of participants on leadership effectiveness, such as “In the last completed project, my leader put my best interests ahead of their own.”
Employee resilience was measured using 5 items developed by Al-Hawari et al. (2020) to underline the importance of personal resource which enables employees to exhibit resilient behaviors when facing challenges. A sample item is “I can get through difficult times at work because I have experienced before difficulties.” The value of Cronbach’s alpha for employee resilience is 0.891.
Organizational support was measured using a 4-item scale developed by Cheng et al. (2003). The 4-item scale of organizational support can effectively examine employee outcomes in workplace, such as job satisfaction, work engagement and turnover intention. The items mainly described the perceived support from their organizations. The sample items include “My organization won't take advantage of me whenever there's an opportunity.” The value of Cronbach’s alpha for organizational support was 0.878.
Work engagement was a 9-item scale developed by Schaufeli et al. (2006). The 9-item scale has good psychometric traits and can be applied to examine employee engagement for positive organizational behavior research, which has been widely used in project context (Nauman et al., 2022; Xia et al., 2022). A sample item is described as “I am immersed in my work.” The Cronbach’s alpha was 0.937.
Reference:
Al-Hawari, M.A.; Bani-Melhem, S.; Quratulain, S. (2020). Do Frontline Employees Cope Effectively with Abusive Supervision and Customer Incivility? Testing the Effect of Employee Resilience. Journal of Business and Psychology, 35, 223–240.
Cheng, B.; Jiang, D.; Riley, J.H. (2003). Organizational Commitment, Supervisory Commitment, and Employee Outcomes in the Chinese Context: Proximal Hypothesis or Global Hypothesis? Journal of Organizational Behavior, 24, 313–334.
Nauman, S., Musawir, A. U., Malik, S. Z., & Munir, H. (2022). Servant leadership and project success: Unleashing the missing links of work engagement, project work withdrawal, and project identification. Project Management Journal, 53(3), 257-276.
Schaufeli, W.B.; Bakker, A.B.; Salanova, M. (2006). The Measurement of Work Engagement With a Short Questionnaire: A Cross-National Study. Educational and Psychological Measurement, 66, 701–716.
Xia, N., Sun, N., & Ding, S. (2022). How psychological capital drives the initiative of project managers in the Chinese construction industry: The roles of work engagement and decision authority. Journal of Management in Engineering, 38(4), 04022031.
3. Industries differ, e.g. construction and IT. Did this affect the results? It is worth investigating and describing this in the paper.
Response: Thanks for your comments. The authors selected these industries because the targeted respondents from these industries in which “PBO form is prevalently utilized to conduct business and production activities” (Sun et al., 2020). (In the sub-chapter 3.1. Sample and Procedures)
And the authors further conducted ANOVA to compare each variable among the different industries. The results are as follows:“Besides, of the respondents, 24.71% were from construction, 33.64% were from information technology, 29.29% were from manufacturing, and the remaining 12.36% were from others. The One-way ANOVA was adopted to compare each variable among respondents engaged in construction, manufacturing, information technology, and other industries. There was no significant difference in servant leadership (F (3, 352)=1.209, p=0.306>0.05), organizational support (F(3, 352)=0.631, p=0.596>0.05), employee resilience (F (3, 352)=0.320, p=0.811>0.05), and work engagement (F(3, 352)=0.639, p=0.590>0.05). Therefore, the One-way ANOVA did not reveal any statistically significant differences in the findings.” (In the sub-chapter 3.1. Sample and Procedures)
Reference:
Sun, X., Zhu, F., Sun, M., Müller, R., and Yu, M. (2020), “Facilitating efficiency and flexibility ambidexterity in project-based organizations: An exploratory study of organizational antecedents”, Project Management Journal, Vol. 51 No. 5, pp. 556-572.

Reviewer 2 Report
Comments and Suggestions for Authors
Thank you for the opportunity to review this article. I found it quite interesting with a clear structure and well-written. Nonetheless, I have some comments:
- Authors should clarify the meaning of the acronym PBO (Project-Based Organization) the first time they mention it in the abstract and in the article. It is explained the second time we see it in the abstract, and it is never explained within the rest of the article.
- In sub-chapter 2.2. authors explain the functions of "servant leadership". Nonetheless, I believe that they should provide a definition of it.
- The same stand for sub-chapter 2.3 and "organizational support". Additionally the paper needs some proofreading (e.g. "Organizational support is a significant job resource that demonstrates organizational support...")
- Regarding "3.1. Sample and procedures": I think that it would be important to know the type of contract that participants have with their employer.
-Regarding "3.2. Measurements": More detailed description of the instruments is needed.
- Figures 2 and 3 (that are mentioned in the results) are missing (unless authors mean figure 1)
Comments on the Quality of English Language
Minor proofreading needed.
Author Response
Reviewer 2 comments
Thank you for the opportunity to review this article. I found it quite interesting with a clear structure and well-written. Nonetheless, I have some comments:
Response: Thanks for the Reviewer’s recognition of this study. The authors especially thank the Reviewer for the invaluable comments, which helped to greatly improve the previous submission.
1. Authors should clarify the meaning of the acronym PBO (Project-Based Organization) the first time they mention it in the abstract and in the article. It is explained the second time we see it in the abstract, and it is never explained within the rest of the article.
Response: Thank you for your comments.
The PBO is the acronym of “project-based organization” in this article. We have clarified it when it is first mentioned in the Abstract section. In the rest of the article, the PBO represents the acronym of project-based organization as well.
2. In sub-chapter 2.2. authors explain the functions of "servant leadership". Nonetheless, I believe that they should provide a definition of it.
Response: Thank you for your suggestions. The definition of servant leadership is described as follows:
“Servant leadership style refers to leaders whose primary purpose for leading is to serve others by investing in their development and well-being for the benefit of completing tasks and objectives for a common goal.” (In the sub-chapter 2.2. section)
Reference:
Eva, N., Robin, M., Sendjaya, S., Van Dierendonck, D., & Liden, R. C. (2019). Servant leadership: A systematic review and call for future research. The leadership quarterly, 30(1), 111-132.
3. The same stand for sub-chapter 2.3 and "organizational support". Additionally the paper needs some proofreading (e.g. "Organizational support is a significant job resource that demonstrates organizational support...")
Response: Thank you for your suggestions. The authors have proofread the sentence as follow:
“Organizational support is an important work resource that reflects the organizational concern for the contributions, welfare and interests of employees (Rhoades and Eisenberger, 2002).” (In the sub-chapter 2.3. section)
Reference:
Rhoades, L.; Eisenberger, R. (2002). Perceived Organizational Support: A Review of the Literature. Journal of applied psychology, 87, 698.
4. Regarding "3.1. Sample and procedures": I think that it would be important to know the type of contract that participants have with their employer.
Response: Thank you for your suggestions.
The authors first illustrate the structure of PBO that participants engaged in. “In this survey, PBO is a kind of permanent organization containing multiple temporary units (projects) (Miterev et al., 2017). Their engaged PBOs conduct the majority of their activities in project mode with some functional support and coordination (Turner et al., 2019).” (In the sub-chapter 3.1. section)
Then the authors illustrate the type of contract that participants have with their employer. “Although the respondents are moving from one project to another or from project mode to organizational roles, they are mostly affiliated to the organizations with permanent employment contracts (Melkonian et al., 2011; Samimi et al., 2021). (In the sub-chapter 3.1. section)
Reference:
Melkonian, T., & Picq, T. (2011). Building project capabilities in PBOs: Lessons from the French special forces. International Journal of Project Management, 29(4), 455-467.
Miterev, M., Mancini, M., and Turner, R. (2017). Towards a design for the project-based organization. International Journal of Project Management, 35(3), 479-491.
Samimi, E., & Sydow, J. (2021). Human resource management in project-based organizations: revisiting the permanency assumption. The International Journal of Human Resource Management, 32(1), 49-83.
Turner, R., and Miterev, M. (2019). The organizational design of the project-based organization, Project Management Journal, 50(4), 487-498.
5. Regarding "3.2. Measurements": More detailed description of the instruments is needed.
Response: Thank you for your comments.
The authors describe the details of the resource of the questionnaire items and the reasons to choose them as follows, which can also be found in the sub-chapter “3.2. Measurements” section.
Servant leadership was measured by a 7-item scale from Nauman et al. (2022). In the research of Nauman et al. (2022), servant leadership functions in a project context to serve the team and motivate a collective synergy for goal attainment. The Cronbach’s alpha was 0.903. The items involve the perception of participants on leadership effectiveness, such as “In the last completed project, my leader put my best interests ahead of their own.”
Employee resilience was measured using 5 items developed by Al-Hawari et al. (2020) to underline the importance of personal resource which enables employees to exhibit resilient behaviors when facing challenges. A sample item is “I can get through difficult times at work because I have experienced before difficulties.” The value of Cronbach’s alpha for employee resilience is 0.891.
Organizational support was measured using a 4-item scale developed by Cheng et al. (2003). The 4-item scale of organizational support can effectively examine employee outcomes in workplace, such as job satisfaction, work engagement and turnover intention. The items mainly described the perceived support from their organizations. The sample items include “My organization won't take advantage of me whenever there's an opportunity.” The value of Cronbach’s alpha for organizational support was 0.878.
Work engagement was a 9-item scale developed by Schaufeli et al. (2006). The 9-item scale has good psychometric traits and can be applied to examine employee engagement for positive organizational behavior research, which has been widely used in project context (Nauman et al., 2022; Xia et al., 2022). A sample item is described as “I am immersed in my work.” The Cronbach’s alpha was 0.937.
Reference:
Al-Hawari, M.A.; Bani-Melhem, S.; Quratulain, S. (2020). Do Frontline Employees Cope Effectively with Abusive Supervision and Customer Incivility? Testing the Effect of Employee Resilience. Journal of Business and Psychology, 35, 223–240.
Cheng, B.; Jiang, D.; Riley, J.H. (2003). Organizational Commitment, Supervisory Commitment, and Employee Outcomes in the Chinese Context: Proximal Hypothesis or Global Hypothesis? Journal of Organizational Behavior, 24, 313–334.
Nauman, S., Musawir, A. U., Malik, S. Z., & Munir, H. (2022). Servant leadership and project success: Unleashing the missing links of work engagement, project work withdrawal, and project identification. Project Management Journal, 53(3), 257-276.
Schaufeli, W.B.; Bakker, A.B.; Salanova, M. (2006). The Measurement of Work Engagement With a Short Questionnaire: A Cross-National Study. Educational and Psychological Measurement, 66, 701–716.
Xia, N., Sun, N., & Ding, S. (2022). How psychological capital drives the initiative of project managers in the Chinese construction industry: The roles of work engagement and decision authority. Journal of Management in Engineering, 38(4), 04022031.
6. Figures 2 and 3 (that are mentioned in the results) are missing (unless authors mean figure 1)
Response: Sorry for the authors’ carelessness. The figure 2 and 3 mean figure 1. The authors have revised figure 2 and 3 in to figure 1 in the sub-chapter “4.3. Structural model” section.
7. Minor proofreading needed.
Response: Thank you for your comments. The authors have checked English language throughout the paper with the assistance of a professional English proof reader. The authors have also referred to some books on the use of academic vocabulary and academic writing in order to make sure the language used in the paper meet the technical standards. Although minor errors may still exist, the authors will devote every effort to correct them.

Reviewer 3 Report
Comments and Suggestions for Authors
The article complies with the publication rules and has attributes that justify its acceptance for publication. However, there are some aspects that require correction:
1) In general, acronyms do not have a plural, so throughout the text PBO (Project-Based Organizations) should be used, and PBOs should be avoided (On lines 252 and 254, the same acronym is mentioned correctly);
2) Between lines 90 and 93, the author makes a textual quotation from Schaufeli et al. (2002), using a portion of text in quotation marks. The page should be indicated.
3) On line 120, there is a comma (,) which should be replaced by a period (.), after the reference mark (31) and before the expression "Resources can be...";
4) On line 122, the mention of the author Bakker includes his first names, so it should be corrected (where it reads: Bakker, Arnold B. et al. (2007), should read Bakker et al. (2007));
5) In line 124, there is a space between "Thus," and "due to the...";
6) In line 225, there is a reference to an author with two names, which should be corrected (as well as in the final references): "Puspo Wiroko. et al. (2021)", which should be changed to "Wiroko et al. (2021)", with the deletion of the excess period (.);
7) On line 289 of the "3.1 Sample and Procedures" section, reference is made to the elimination of 81 questionnaires. Because this is a very high number, I suggest clarifying the reasons for eliminating such a large number of responses;
8) There is an excess comma in line 301, corresponding to Schaufeli et al. (2006);
9) In lines 347 and 353, reference is made to Figures 2 and 3 respectively. Since there are no Figures 2 and 3 included in the text, this inconsistency should be rectified, in one sense or another.
From the point of view of the substance of the body of the article, section "6. Implications" somewhat repeats what has already been stated and section "7. Limitations and Future Research" doesn't seem to add anything relevant in terms of limitations and suggestions for future studies.
In general, the article deserves to be published.
Author Response
Reviewer 3 comments
The article complies with the publication rules and has attributes that justify its acceptance for publication. However, there are some aspects that require correction:
1. In general, acronyms do not have a plural, so throughout the text PBO (Project-Based Organizations) should be used, and PBOs should be avoided (On lines 252 and 254, the same acronym is mentioned correctly);
Response: Thanks very much. According to your suggestions, the authors have used the “PBO” as the acronym of the Project-Based Organizations through the manuscript.
2. Between lines 90 and 93, the author makes a textual quotation from Schaufeli et al. (2002), using a portion of text in quotation marks. The page should be indicated.
Response: Thank you for your suggestions. The textual is quoted from the 74 page of Schaufeli et al. (2002). The page 74 has added in the text as follows.
Furthermore, Schaufeli et al. (2002) defined work engagement as “a positive, fulfilling, work-related state manifested by vigor, dedication, and absorption in work roles” (p.74). (In the sub-chapter 2.1. section)
Reference:
Schaufeli, W.B.; Salanova, M.; González-Romá, V.; Bakker, A.B. (2002). The Measurement of Engagement and Burnout: A Two Sample Confirmatory Factor Analytic Approach. Journal of Happiness studies, 3, p.74.
3. On line 120, there is a comma (,) which should be replaced by a period (.), after the reference mark (31) and before the expression "Resources can be...";
Response: Thank you for your suggestions. The authors have replaced the comma (,) with a period (.) (In the sub-chapter 2.1. section)
4. On line 122, the mention of the author Bakker includes his first names, so it should be corrected (where it reads: Bakker, Arnold B. et al. (2007), should read Bakker et al. (2007));
Response: Thank you for your suggestions. The authors have revised “Bakker, Arnold B. et al. (2007)” as “Bakker et al. (2007)”. (In the sub-chapter 2.1. section)
5. In line 124, there is a space between "Thus," and "due to the...";
Response: Thank you for your suggestions. The space was added between "Thus," and "due to the..." (In the sub-chapter 2.1. section)
6. In line 225, there is a reference to an author with two names, which should be corrected (as well as in the final references): "Puspo Wiroko. et al. (2021)", which should be changed to "Wiroko et al. (2021)", with the deletion of the excess period (.);
Response: Thank you for your suggestions. The authors have changed "Puspo Wiroko. et al. (2021)" to "Wiroko et al. (2021)" and deleted the excess period (.) as well. (In the sub-chapter 2.5. section)
7. On line 289 of the "3.1 Sample and Procedures" section, reference is made to the elimination of 81 questionnaires. Because this is a very high number, I suggest clarifying the reasons for eliminating such a large number of responses;
Response: Thank you for your suggestions.
Besides the invalid questionnaires answered in a short time, the questionnaires which were not answered by the participants that engaged in our required PBO form required were invalid as well. This is mainly because our target participants engaged in PBO which is a permanent organization containing multiple temporary units (projects) with permanent employment contracts. However, some of the participants engaged in pure temporary organizations. This may increase the number of elimination of questionnaires. The authors have illustrated the reasons as follows:
“Therein, some of the questionnaires answered in a short time were invalid. Besides, some of the questionnaires which were not answered by the participants that engaged in the PBO form required by this study were invalid as well.” (In the sub-chapter 3.1. section)
8. There is an excess comma in line 301, corresponding to Schaufeli et al. (2006);
Response: Thank you for your suggestions. The authors have added the excess comma between Schaufeli et al. and (2006). (In the sub-chapter 3.2. section)
9. In lines 347 and 353, reference is made to Figures 2 and 3 respectively. Since there are no Figures 2 and 3 included in the text, this inconsistency should be rectified, in one sense or another.
Response: Sorry for the authors’ carelessness. The figure 2 and 3 mean figure 1. The authors have revised figure 2 and 3 in to figure 1 in the sub-chapter “4.3. Structural model” section.
10. From the point of view of the substance of the body of the article, section "6. Implications" somewhat repeats what has already been stated and section "7. Limitations and Future Research" doesn't seem to add anything relevant in terms of limitations and suggestions for future studies.
Response: Thank you for your suggestions. The authors have restructured the section 6 and section 7.
(1) In the “6. Implications” section
The first paragraph describes the theoretical implications. “This study explores how servant leadership motivates positive work behaviors by providing employees with different work resources. This study extends the results of servant leadership, particularly in PBO. Considering the multiple work roles and work stress that eroding employee well-being, this study underlines the significance of people-centered of servant leadership in the project management practices of PBO. By clearly defining how servant leadership functions in PBO, this study proposes an effective and positive management intervention to prioritize values, support, and care. Furthermore, based on COR and JD-R theory, this study identifies two urgent work-related resources by which project leaders influence positive work states and behaviors of employees. The two resources are employee resilience (personal resources) and organizational support (job resources). Both of the work resources emerge as vital mediator between servant leadership and work engagement. The findings extend the research for antecedents of work engagement and explain how to enhance work engagement effectively through these two work resources.”
Then, the rest paragraphs provide the practical implications.
For example, the second paragraph suggests that PBO should pay attention servant leadership by selecting and training service orientation managers and employees. “……Organizations recognize the critical role of servant leaders in motivating work engagement. Therefore, PBO focuses on providing people-oriented training to develop and promote service behaviors in project managers. In addition, when hiring project managers and project team members, PBO should gauge the service orientation of candidates.”
The third paragraphs provide implications for project leaders to enhance work-related resources by providing organizational support. “……Thus, project managers concentrate on demonstrating such empowering, supportive, and people-oriented behaviors for the duration of the project. For instance, they can offer organizational support to employees across various dimensions, including the human environment, work system processes, compensation, and benefits. Developing a relatively comfortable and convenient work environment, alongside designating clear performance appraisal standards, ensures that PBO employees are fully engaged in their work. …. As a result, employees with higher perception of organizational support actively reduce the negative impact of resource scarcity on their work engagement through a positive mindset, which is key to enhancing sustainable employee well-being.”
The fourth paragraph describes how to improve employee resilience for project leaders. “Examples include cultivating employee resilience by enhancing their professional knowledge and competencies to cope with stressful work environments. Simultaneously, greater attention should be given to developing employee resilience to bolster their psychological capital against work pressure. In addition, the implementation of employee assistance programs, targeted solution models, positive thinking reduction therapy, short-term introspection therapy, and therapeutic introductions can foster positive hope and optimism, encouraging individuals to concentrate on their work and cultivate a higher level of work dedication.”
(2) In the “7. Limitations and Future Research” section
The first paragraph describes the limitations for research design. On one hand, “The sample utilized in this research was confined to a single country, China, potentially limiting the generalizability of our results across cultural contexts. Future research could validate the findings with more diverse and cross-cultural samples.” On the other hand, “The purpose of this cross-sectional study is to explore the mechanisms by which servant leadership influences work engagement in PBO, but it is not clear whether there is a causal relationship between each variable. The relationship among different variables is not a direct causal relationship but is associated with a certain causal sequence, which needs to be further confirmed in prospective long-term studies with a time-lag design.”
The second paragraph describes the direction for future Research. Although this study identifies two urgent work resources for PBO to enhance work engagement, these may be inadequate. Maybe, other work resources are effective. Regarding this gap, this study further proposes social capital (organizational resource) and self-efficacy (personal resource) for future research, which can be found as follows:
“Although organizational support and employee resilience are identified as urgent work resources in PBO, other types of significant resources are suggested to enhance employee work engagement in the future. For example, social capital is an organizational resource embedded in the relational network (Zhang and Cheng, 2015). And self-efficacy is another vital personal resource that enables employees to carry out tasks to realize specific objectives (Namaziandost et al., 2023). Therefore, future research is suggested to examine the important roles of those work resources in improving work engagement in PBO.”
Namaziandost, E., Heydarnejad, T., Rahmani Doqaruni, V., & Azizi, Z. (2023). Modeling the contributions of EFL university professors’ emotion regulation to self-efficacy, work engagement, and anger. Current Psychology, 42(3), 2279-2293.
Zhang, L., & Cheng, J. (2015). Effect of knowledge leadership on knowledge sharing in engineering project design teams: the role of social capital. Project Management Journal, 46(5), 111-124.
11. In general, the article deserves to be published.
Response: Thanks for the Reviewer’s recognition of this study. The authors especially thank the Reviewer for the invaluable comments, which helped to greatly improve the previous submission.
